# Impact of essential workers in the context of social distancing for epidemic control

**William R. Milligan**[1]*, **Zachary L. Fuller**[1], **Ipsita Agarwal**[1], **Michael B. Eisen**[2,3]*, **Molly Przeworski**[1,4,5], **Guy Sella**[1,5]*

**1** Department of Biological Sciences, Columbia University, New York City, New York, United States of America, **2** Howard Hughes Medical Institute, University of California, Berkeley, California, United States of America, **3** Department of Molecular and Cell Biology, University of California, Berkeley, California, United States of America, **4** Department of Systems Biology, Columbia University, New York City, New York, United States of America, **5** Program for Mathematical Genomics, Columbia University, New York City, New York, United States of America

* wm2377@columbia.edu (WRM); mbeisen@berkeley.edu (MBE); gs2747@columbia.edu (GS)

## Abstract

New emerging infectious diseases are identified every year, a subset of which become global pandemics like COVID-19. In the case of COVID-19, many governments have responded to the ongoing pandemic by imposing social policies that restrict contacts outside of the home, resulting in a large fraction of the workforce either working from home or not working. To ensure essential services, however, a substantial number of workers are not subject to these limitations, and maintain many of their pre-intervention contacts. To explore how contacts among such "essential" workers, and between essential workers and the rest of the population, impact disease risk and the effectiveness of pandemic control, we evaluated several mathematical models of essential worker contacts within a standard epidemiology framework. The models were designed to correspond to key characteristics of cashiers, factory employees, and healthcare workers. We find in all three models that essential workers are at substantially elevated risk of infection compared to the rest of the population, as has been documented, and that increasing the numbers of essential workers necessitates the imposition of more stringent controls on contacts among the rest of the population to manage the pandemic. Importantly, however, different archetypes of essential workers differ in both their individual probability of infection and impact on the broader pandemic dynamics, highlighting the need to understand and target intervention for the specific risks faced by different groups of essential workers. These findings, especially in light of the massive human costs of the current COVID-19 pandemic, indicate that contingency plans for future epidemics should account for the impacts of essential workers on disease spread.

## Introduction

New emerging infectious diseases are identified every year [1], a subset of which become global pandemics (e.g., COVID-19, H1N1, HIV, and Zika). In the past 20 years alone, several viral respiratory diseases have emerged [2], including three resulting from novel coronaviruses

**Data Availability Statement:** All data and code used to generate figures is available on Github (https://github.com/zfuller5280/Covid-19_Modeling/).

**Funding:** This work was supported by funding from the National Science Foundation (DGE

1644869 to WM, www.nsf.gov). The funders had no role in study design, data collection and analysis, decision to publish, or preparation of the manuscript.

**Competing interests:** The authors have declared that no competing interests exist.

(SARS, MERS, and COVID-19), many of which required public health interventions to prevent disease transmission [3–5]. In the case of COVID-19, these interventions often involve some form of "shelter in place" (SIP) in which the majority of the population remains in their homes except for essential activities like grocery shopping. The motivation is to either locally eradicate the infection, or to reduce its spread enough to decrease peak demand on healthcare and gain time to develop testing capacity, therapies and vaccines. SIP orders have been guided by extensive modeling of the COVID-19 pandemic to predict its future and understand its impact on the population under various scenarios, including different stringencies of SIP (e.g., [6–8]).

By necessity, SIP involves exceptions for "essential workers", typically including those involved in the delivery of health care, the production and distribution of food, emergency services and defense, public works and utilities, communications and information technology, and logistics and delivery. The fraction of workers designated as essential varies geographically due to different regulations and the makeup of the local economy [9]. Within the United States, industries designated as essential are estimated to employ approximately 40% of the workforce [9]. In New York City, workers in categories deemed essential (as of March 2020 [10]) are estimated to comprise a quarter of the workforce [11], or over 1M people, of whom over half are employed in healthcare and 15% in grocery, convenience and drug stores. Estimates in California are that one in eight individuals is considered an essential worker [12].

The sheer number of essential workers, and their exemptions during SIP, suggests they may have unique and substantial impacts on epidemic control. By necessity, essential workers maintain many of their contacts [13, 14], which puts them at both increased risk of becoming infected and of potentially infecting others [15–22]. Moreover, different types of essential workers differ in their contacts among themselves and with others, which suggests their risks of infection and of infecting others plausibly differ as well. Yet while healthcare workers have received a great deal of attention, for obvious reasons, other kinds of essential workers much less so. Despite their likely impact on epidemic dynamics, most models of the COVID-19 pandemic, including those used to guide policy [23–28] do not explicitly consider essential workers, let alone differences among them. Thus, we lack an understanding of the impacts that essential workers have on the spread of the pandemic and of policy measures that could ameliorate these impacts.

As a step toward closing this gap, here, we extend the widely-employed "SEIR" epidemiological model [6–8] to qualitatively evaluate the individual infection risk faced by different types of essential workers, and the impact that essential workers have on pandemic control. Similar to models that compartmentalize the population by location (e.g., zip code or county [23, 29]) or by age [26, 28], we explicitly model transmission within and between two subpopulations: essential workers and everyone else. We focus on the impact of essential workers on the spread of COVID-19, but our results should generalize to any infectious disease where SIP is used to prevent transmission.

## Modeling essential workers

We began by implementing a standard SEIR model to describe the dynamics of COVID-19 in a population (Fig 1A). Following previous work [6–8], we included three types of infected individuals, corresponding to those destined to a) show no symptoms, b) have symptoms but not require hospitalization, and c) require hospitalization. We also included three types of hospitalized individuals corresponding to those who will a) recover without critical care, b) require critical care but will recover, and c) die after receiving critical care. We make the standard, simplifying modeling assumption that recovered individuals cannot be reinfected.

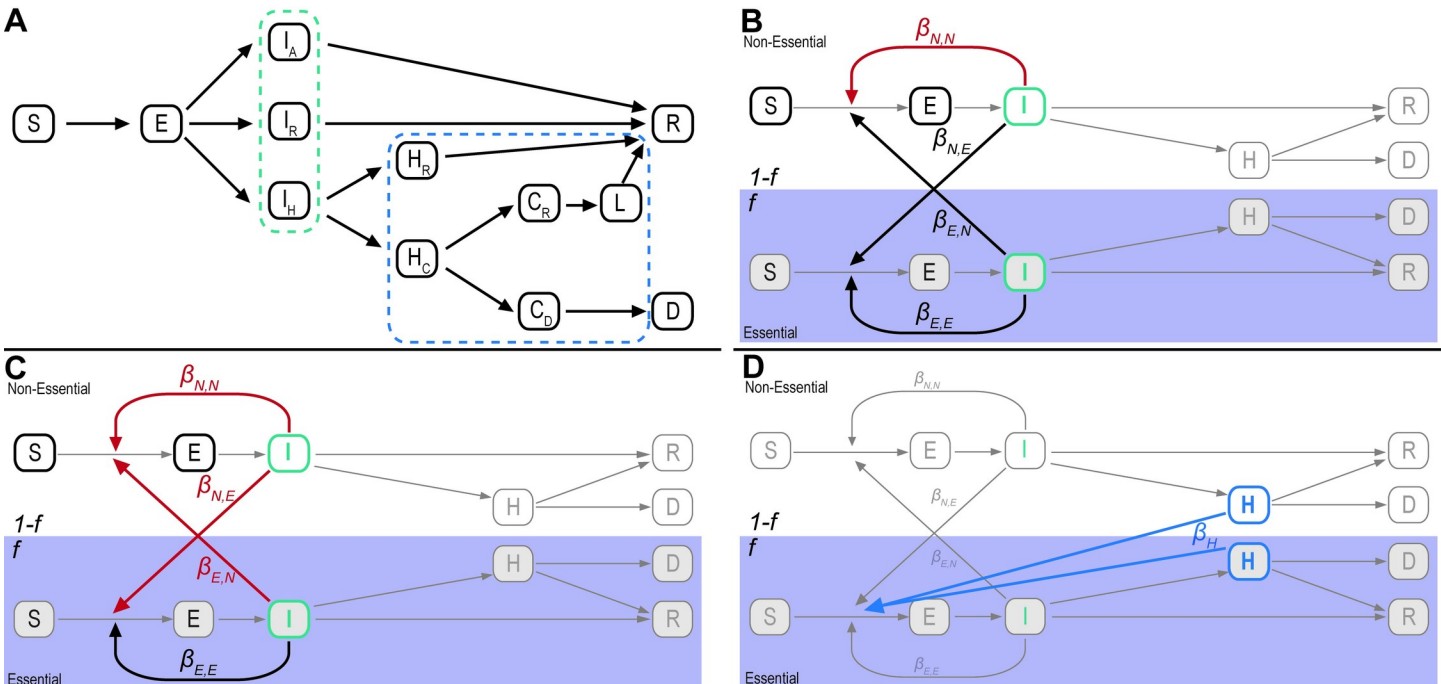

**Fig 1. Diagram of the SEIR model with extensions. A)** An illustration of the basic SEIR model used in all scenarios, including additional compartments within the infected (outlined in green) and hospitalized (outlined in blue) classes. 'S' is susceptible, 'E' is exposed, 'I' is infected, 'H' is hospitalized, 'D' is dead, and 'R' is recovered. Within the infected class, individuals can be asymptomatic '$I_A$' and destined to recover; symptomatic '$I_R$' but destined to recover; or symptomatic '$I_H$' and destined to be hospitalized. Within the hospital, individuals either go to recovery '$H_R$' or go to critical care '$H_c$'. For those in critical care, individuals either die '$C_D$' or go on to the recovered class '$C_R$', with an additional time spent in the hospital 'L'. **B)** To model the impact of EWs, who make up a proportion $f$ of the total population, we created two cloned instances of the SEIR model. Here, for visualization, the infectious and hospitalized classes are collapsed into a single compartment. The $\beta$ terms represent the transmission routes between infectious individuals within and between essential 'E' and non-essential 'N' groups. In the model of public-facing EWs (such as cashiers, transportation workers and public safety personnel), SIP reduces contacts (highlighted in red) only among nEWs. **C)** In the model of non-public-facing EWs (such as factory, warehouse, and agricultural workers), SIP reduces all contacts (highlighted in red) except for those among other EWs. This model is relevant for EWs that can social distance from nEWs but not from each other. **D)** To model the impact of healthcare workers, an additional infectious route ($\beta_H$) is included from within the hospitalized compartments to susceptible individuals in the essential group.

We parameterized the differential equations that describe the rate at which individuals transition between model states based on the epidemiological literature for COVID-19 (see Table 1), and chose the lockdown date based roughly on New York City (see Modeling Details). Although it was not our goal to model the outbreak in any particular region, we verified that the parameters used match observed pandemic dynamics during March and April of 2020 in three US cities (see Section 5 Fitting to empirical data of S1 Appendix). In S1 Appendix, we vary parameter values to check that our qualitative results are insensitive to variation within the range examined (See Figs 1–16 in S1 Appendix for alternative versions of Fig 2, Figs 17–29 in S1 Appendix for Fig 3, and Figs 20–46 in S1 Appendix for Fig 4).

We next created two cloned instances of the SEIR model, one for essential workers and one for everyone else, and connected them via terms that describe the probability that an infected individual in one subpopulation infects a susceptible individual in the other (Fig 1B–1D). In what follows, we refer to these two subpopulations as essential workers (EWs) and non-essential-workers (nEWs), the latter category encompassing all other workers and people not in the labor force. The models for EWs and nEWs have identical structure and parameters, except for the within and between subpopulation transmission rates. We do not attempt to fit the model including EWs due to lack of longitudinal data on infections differentiating EWs from nEWs (see Discussion).

**Table 1. Parameter estimates for the model.**

| Parameter | Description | Literature Estimates | Values Used |
|---|---|---|---|
| $R_0$ | | 2–5 [30] | 3 |
| | | 2.4–2.6 [7] | |
| | | 2–7 [31] | |
| | | 1.5–6.5 [32] | |
| $t_E$ | Latency period | 4.6 days [6,7] | 3 days* |
| | | 5 days (1–14) [32–35] | |
| $t_{IA}$ | Infectious period (asymptomatic class) | 5 days [6, 7, 34] | 5 days |
| | | 8 days [33] | |
| $t_{IR}$ | Time of between infection and recovery (mild symptoms) [infectious period for mild cases] | 5 days [6, 7, 36] | 5 days** |
| | | 8 days [33] | |
| | | Median 2 weeks [37] | |
| | | ~12 days [38] | |
| | | < 18 days [39] | |
| | | 1–2 weeks [40] | |
| $t_{IH}$ | Time between infection and hospitalization | 5 days [6, 7, 34] | 5 days |
| | | 1 week [37] | |
| | | 5–12 days [41] | |
| | | 7 +/- 4 days [42] | |
| $t_{HR}$ | Time in hospital (with no critical care) | 8 days [6, 7] | 8 days |
| | | 11 days [33] | |
| **Total duration of infectious period through critical care path (to recovered or dead)** | | 21 days [6, 7] | 21 days |
| | | 3–6 weeks [37] | |
| | | 16–23 days [42] | |
| $t_{HC}$ | Time in hospital before critical care | 6 days [6, 7] | 6 days |
| $t_{CR}$ | Time in critical care before recovery | 10 days [6, 7] | 7 days*** |
| | | Includes $t_{CR}$ and $t_L$ | |
| $t_L$ | Time in hospital after leaving critical care | - | 3 days*** |
| $t_{CD}$ | Time in critical care before death | 10 days [6, 7] | 10 days |
| $p_{EIA}$ | Proportion of infections that are asymptomatic | 33% [6, 7] 17.9% [43] | 33% |
| | | 13.5% [44] 25% [45] | |
| $p_{EIH}$ | Proportion of infections requiring hospitalization | 4.4% [6, 7] | 4.4% |
| $p_{EIR}$ | Proportion of symptomatic not requiring hospitalization | - | 62.6% |
| $p_{IHR}$ | Proportion of hospitalizations that do not require critical care | - | 70% |
| $p_{IHC}$ | Proportion of hospitalizations that require critical care | 30% [6, 7, 46] | 30% |
| | | 26%-32% [41] | |
| $p_{HCR}$ | Proportion of recoveries from critical care | - | 50% |
| $p_{HCD}$ | Proportion of deaths in critical care | 50% [6, 7] | 50% |
| | | 39% to 72% [41] | |

* We reduce the latency period from approximately five days to three days to account for pre-symptomatic transmission [38, 39]. When there is substantial variation in estimates of a parameter, we choose what appears to be the consensus value (e.g., ** $t_{IR}$ = 5 days).

*** We split the estimate of ten days in critical care before recovery into seven days in critical care and three days in the hospital after leaving critical care.

For our analysis, the critical characteristic of EWs that distinguishes them from nEWs is that they maintain a substantial fraction of their work-associated contacts after the institution of SIP. However, there is great diversity among EWs in their contact profiles. For example, factory, warehouse and agricultural workers retain contacts with the other employees at their

place of work, but have their contacts with the remainder of the population reduced by SIP. Others, such as cashiers, transportation workers, and police have frequent contacts with many people who are nEWs. Hospital workers, in turn, are in contact not only with themselves, but also with the people hospitalized with the virus.

We therefore generated three separate EW-containing SEIR models, one for each of these archetypal EWs. These models differ in how an individual's contacts are distributed within and between subgroups, in how SIP affects these contacts, and, for healthcare workers, in which individuals are the source of new infections.

The three models have two shared parameters: $f$, the fraction of the population that are EWs; $\theta$, the remaining proportion of individual to individual disease transmission after social distancing ($\theta = 1$ is no social distancing; $\theta = 0$ is complete isolation for everyone). Without EWs ($f = 0$), the pandemic is suppressed when $R_0\theta$ (the post-SIP number of new infections expected to arise from an infected individual in a fully susceptible population) is less than one.

Mathematical details and full parameter choices for each model are described in the Modeling Details section; the implementation and the results of simulations with these models (as well as variations on them) are available in S1 Appendix.

## Model 1: Public-facing essential workers (Fig 1B)

We began by considering workers such as cashiers and other shopworkers, transportation workers and public safety personnel, whose work involves extensive contact with nEWs. The critical feature of our model of such "public-facing" EWs is that only contacts among nEWs are reduced by SIP.

## Model 2: Non public-facing essential workers (Fig 1C)

Unlike public-facing EWs, factory, warehouse, and agricultural workers interact extensively with other EWs, but their work does not involve contact with nEWs. The critical feature of our model of such "non-public-facing" EWs is that all contacts except those among EWs are reduced by SIP. We further assume that half of non-public-facing EW's contacts are with other essential workers; in contrast, for public-facing essential workers, we assume contacts between subpopulations are symmetric and proportional to subpopulation size. These features are meant to account for frequent and close contacts in the workplace, and may be essential to explaining outbreaks within, say, factories [47, 48].

## Model 3: Healthcare workers (Fig 1D)

Frontline healthcare workers are exposed to infected individuals in hospitals and other critical care settings. We therefore created a specific "healthcare worker" model with an additional interaction term describing the rate at which individuals hospitalized with COVID infect susceptible healthcare workers (see Modeling details). Neither these hospital-specific infections nor infections among healthcare workers are affected by SIP, in contrast to contacts between healthcare workers and nEWs, as well as contacts among nEWs. Similar to the non-public-facing essential worker model, we assume half of healthcare workers' contacts are with other healthcare workers because of their close contact at the workplace (see Figs 10, 23, and 40 in S1 Appendix for results with proportionate mixing). For simplicity, we further assume all COVID-19 patients in any hospital compartment are equally infectious (see Figs 16, 29, and 46 in S1 Appendix for results where only patients in the $H_R$ and $H_C$ compartments are infectious).

## Results

### Essential workers have elevated infection risk

We began by examining the personal infection risk of each class of EW. Fig 2 shows the cumulative fraction of EWs and nEWs infected as the pandemic progresses for each model, with $f$ = **0.05** (5% of the population is EWs) and $\theta$ values corresponding to partially effective ($\theta = 0.5$ and $R_0\theta = 1.5$), effective ($R_0\theta = 0.9$ and $\theta = 0.3$) and highly effective ($R_0\theta = 0.5$ and $\theta = 0.16$) SIP.

For all three models, EWs have a substantially higher risk of infection than nEWs. The risk is greatest for healthcare workers, who, under our model parametrization, are nearly all infected quite rapidly, even when the rest of the population is under stringent SIP. But both public-facing and non-public-facing EWs have much higher risk than nEWs: non-public-facing EWs are susceptible to the wave of infection that sweeps through their workplaces even after SIP, while public-facing EWs are exposed to infected people at a much higher rate than those under SIP.

Our model predicts a somewhat higher infection prevalence over the first 100 days than seems plausible, likely because we are not taking into account measures taken by EWs, in particular healthcare workers, to reduce transmission in their workplaces.

### Public-facing essential workers increase infection risk in the remainder of population

Although the individual risk to public-facing EWs is relatively low, they have a substantial impact on nEWs. In conditions when SIP would be expected to be effective at controlling the growth of the pandemic ($R_0\theta<1$), having 5% of the population working in public-facing EW jobs leads to a nearly 50% increase in the number of nEWs who are infected (Fig 2).

We were initially surprised that the high rate of infection of both non-public-facing and healthcare EWs led only to marginal increases in the infection risk for nEWs (Fig 2). However, the combination of the model assumptions of a relatively small EW subpopulation (5% of the

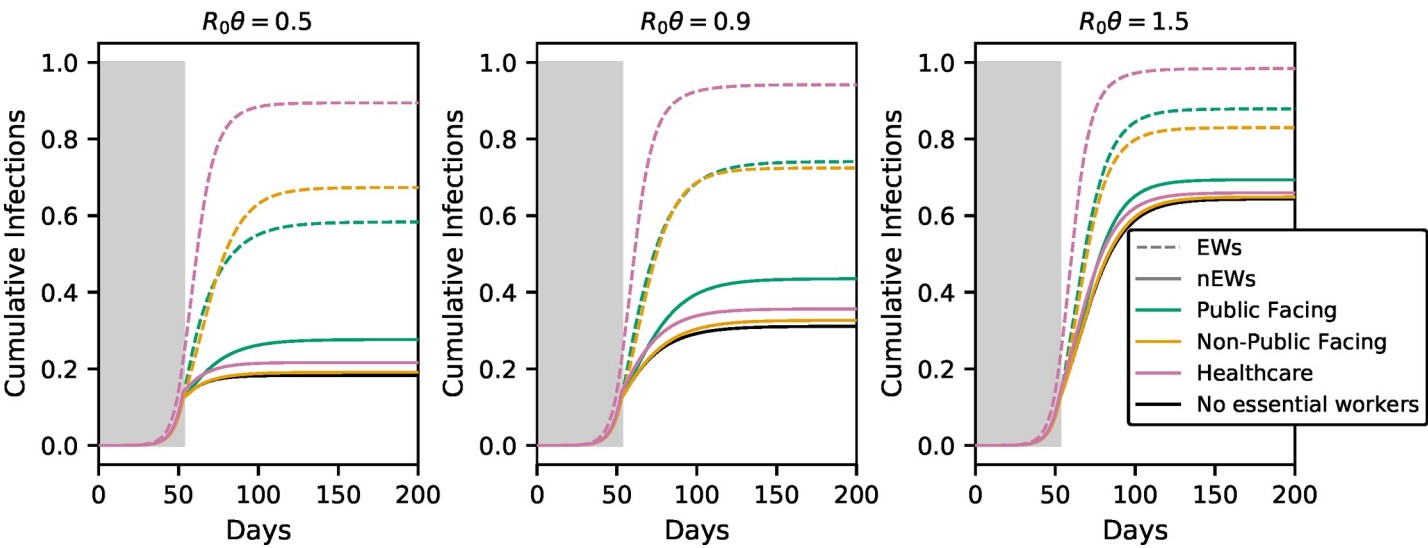

**Fig 2. Cumulative infection rates among EWs and nEWs for different scenarios and values of $\theta$.** The dashed and solid lines correspond to EWs and nEWs respectively. Note that the ordering of the colors is not the same for EWs and nEWs. The proportion of EWs, $f$, is assumed to be 0.05 for all models. Alternative values of $f$ yield similar qualitative results, as do alternative values of other parameters (see Section 2 and Figs 1–16 in S1 Appendix).

total), that half of EW contacts are with other EWs (see Model Details), and SIP suppression of contacts between EWs and the rest of the population means that, even when there is rampant infection among EW, there is a low leakage of infections to nEWs. And, although some transmission from EWs to nEWs does occur, those infections lead to little onward transmission among nEWs, due to the SIP.

### Effects on pandemic control of increasing number of essential workers

The differing effects of EWs on disease risk in EWs and nEWs led us to next examine how the different types of EW impact the pandemic. Fig 3 shows the total fraction of the population expected to be infected after one year as a combined function of $f$ and $\theta$ for all three EW models (Fig 3A), as well as the breakdown for EWs (Fig 3B) and nEWs (Fig 3C). The red contour line in each panel of Fig 3 represents $f$, $\theta$ values for which the total fraction of the population infected after one year is equal to the fraction infected for $f = 0$ and $R_0\theta = 1$. Values below and to the right of this band result in fewer people infected, values above and to the left in more.

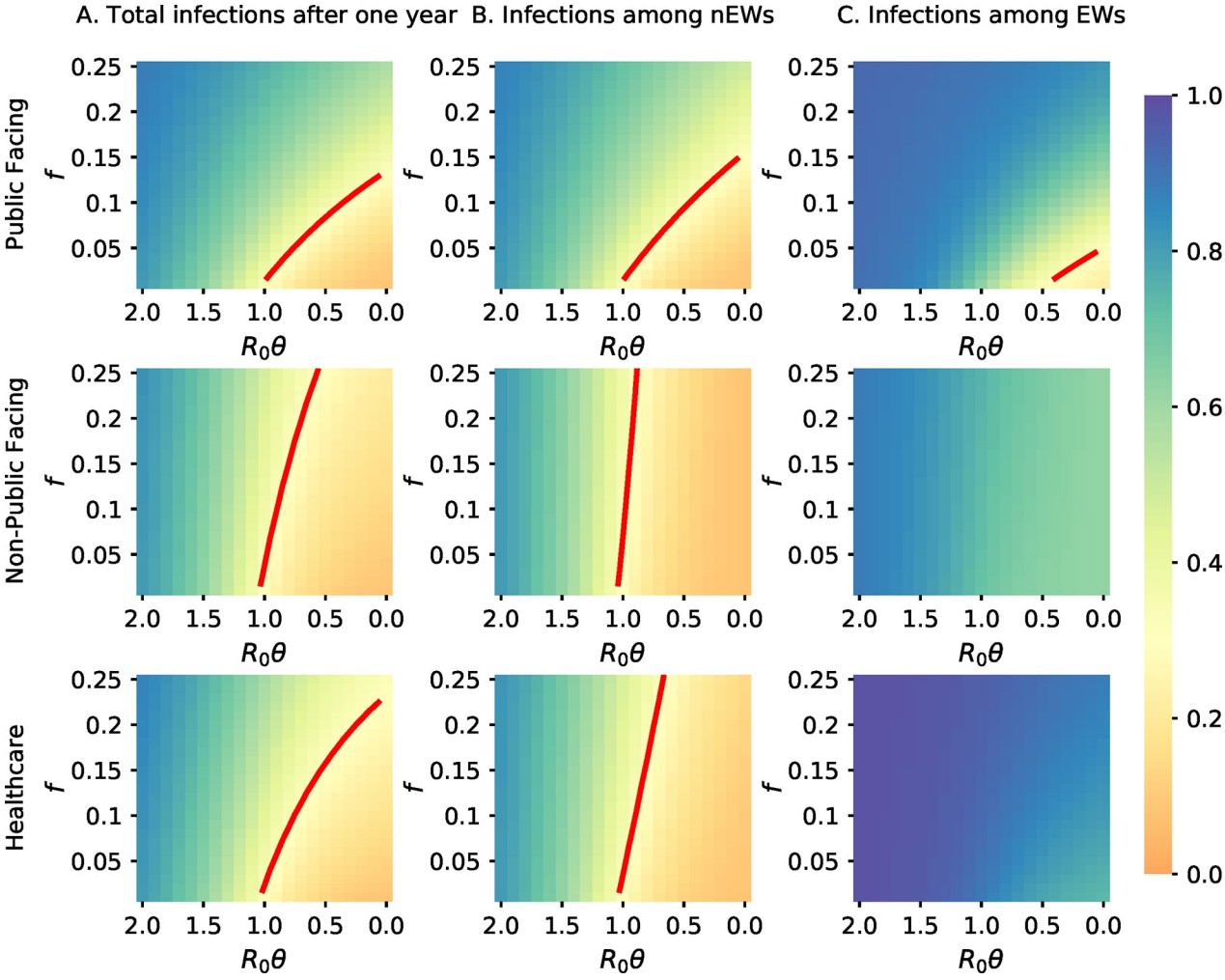

**Fig 3.** Heatmaps of cumulative infections after a year in the total population (A), in nEWs (B), and in EW (C). The red contour lines correspond to $f$, $\theta$ values for which the prevalence of infection over a year in the population or subpopulation is equal to the prevalence without EWs (i.e., $f = 0$) and $R_0\theta = 1$. These contour lines are absent in the two bottom right-most panels, because when $f > 0$, the prevalence in EWs is greater than that expected with $f = 0$ and $R_0\theta = 1$. For equivalent figures with alternative choices of parameter values, see Section 3 and Figs 17–29 in S1 Appendix.

In all models, increasing the number of EWs requires a compensatory increase in the stringency of SIP. However, there is considerable difference in the stringency required in each case. An approximately two-fold greater decrease in $R_0\theta$ is required to compensate for an increase in the number of public-facing EWs compared to the same increase in the number of non-public-facing and healthcare EWs.

## Dynamics of infections to, from and within essential worker subpopulations

The differences in infectious interactions within and between subpopulations after SIP results in a complex dynamics of the number and source of infections of EW and nEW over the course of a year. To investigate their nature, and understand how they manifest in the three different EW models, we examined the prevalence (the number of infected individuals on a given day) of infection in EWs and nEWs assuming $f = 0.05$ and $R_0\theta = 0.5$ as a function of time (Fig 4A).

Before SIP, the epidemic progression is identical in public facing EWs, non-public facing EWs, and nEWs (Fig 4A) and the fraction of infected individuals that are EWs reflect their proportion in the population $f$ (Fig 4B). Healthcare workers, however, bear a proportionally larger burden of the epidemic even before SIP, because they are subject to additional within-hospital infections (Fig 4B).

After SIP is imposed, the prevalence of infections in nEWs begins to decline, but, because EWs cannot social distance as effectively as nEWs, the prevalence in EWs continues to rise, albeit more slowly (Fig 4A). In all three models, infections in EWs rapidly peak, after which EW prevalence decays at the same rate as nEW prevalence, albeit at a higher level. There is, however, a striking difference between the public-facing EW model and the other two, with both a slower decay of prevalence, and less of a gap between EWs and nEWs in the public-facing EW model.

This difference is more evident when examining the fraction of all infections that are in EWs as a function of time (Fig 4B). In the public-facing EW model, there is a rapid rise from the initial setting of 0.05 to a peak of 0.15, where it levels off. In contrast both the non-public facing and healthcare EW models reach a point where roughly half of all infections occur in EWs before stabilizing slowly to a value of approximately one third. This difference arises because under the public-facing worker model, the rate of contact between EWs and nEWs is not reduced by SIP, decreasing the divergence in the rate of infection of the two subpopulations. In contrast, in the non-public facing and healthcare worker models, SIP largely decouples the epidemic of EWs from nEWs. Thus, the increased prevalence among EWs does not cause substantially more infections among nEWs, resulting in EWs comprising a disproportionately large proportion of the infections (i.e., $>>f$).

The changes in the distribution of infections across the EW and nEW subpopulations over time results in complex, shifting patterns of who is infecting who (Fig 4C). In all models, before SIP, most infections spread among nEWs, as they constitute the vast majority of the population. Immediately after SIP, infections stemming from contacts that are reduced by SIP drop sharply (e.g., EN in non-public facing and NN in all models) and conversely, the proportion of infections from contacts unaffected by SIP increases. In the public-facing EW model, there is a rapid shift post-SIP to there being a roughly equal probability that a new nEW infection came from either another nEW or an EW—a striking result given that only 5% of the population are EWs. In contrast, public-facing EWs are roughly ten times more likely to be infected by a nEW than an EW.

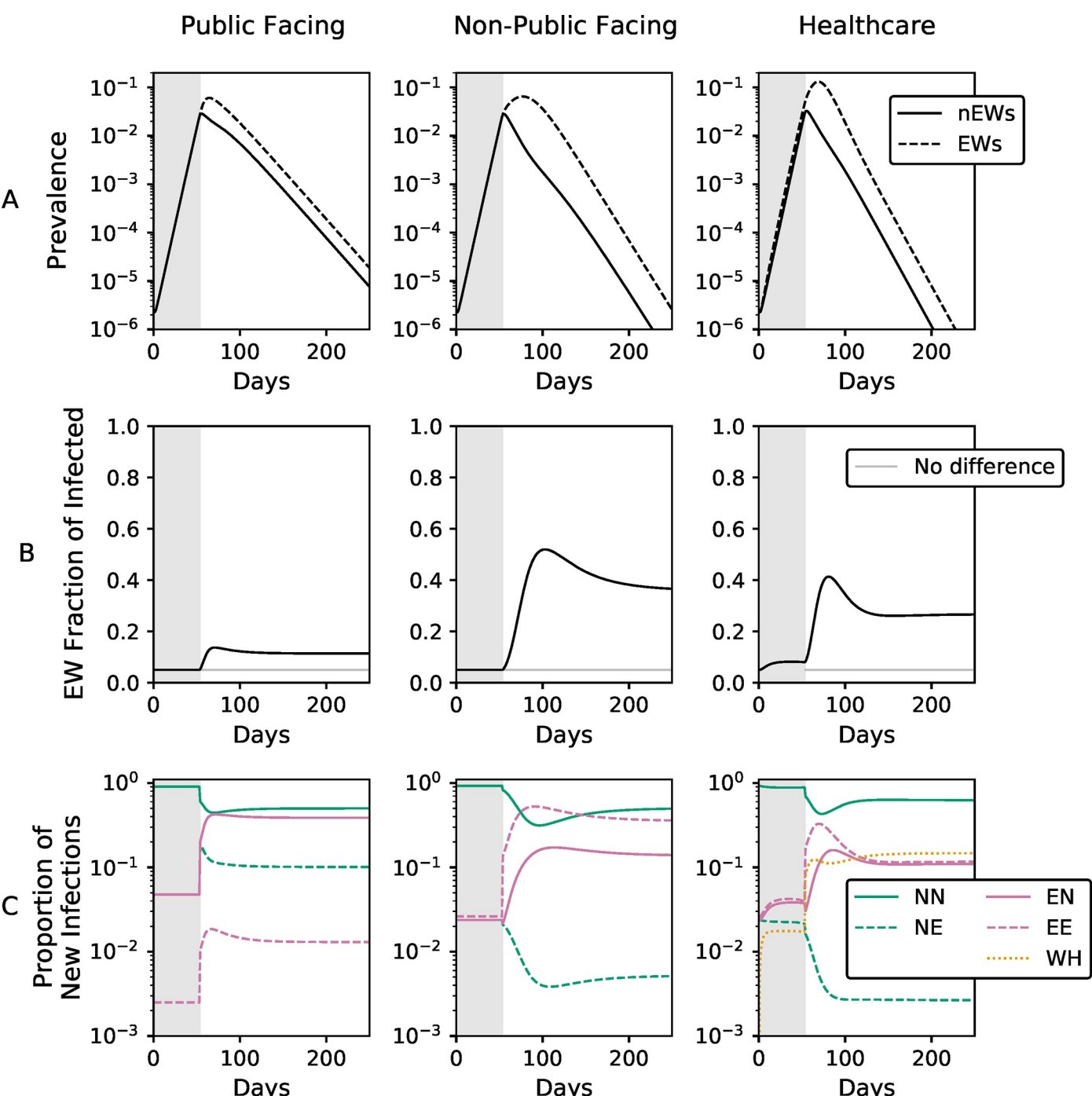

**Fig 4. Time resolved dynamics of infections, for $f = 0.05$ and $R_0\theta = 0.5$.** Times before the implementation of SIP (day 53) are denoted by the grey shade. **A)** Time-resolved proportion of EWs (dashed lines) and nEWs (solid lines) that are infected. **B)** The fraction of infected individuals that are EWs. Prior to SIP, this fraction is $f$, except in the healthcare model where EWs also become infected from individuals within the hospital compartments. After SIP, this fraction increases, *i.e.*, EWs bear a proportionally larger burden of the epidemic. **C)** Where new infections originate. SIP reorganizes the flow of infections through the subpopulations in a model-dependent manner. The acronyms denote different types of transmission, with the first and second letters denoting the infecting and the infected subpopulation respectively (e.g., EN is infections from EWs to nEWs); WH refers to infections that occur within hospitals. For equivalent figures with alternative choices of parameter values, see Section 4 and Figs 30–46 in S1 Appendix.

The non-public-facing EW model has quite different dynamics. It takes longer for EWs to become a significant source of new infections for nEWs, a product of the time it takes for the infection to spread extensively within the EW subpopulation. In contrast to the public-facing model, where EW to nEW and nEW to nEW are effectively tied for the most common type of infection transfer, in the non-public-facing model, the two most common infection types are both within group terms. The dynamics of the healthcare EW model are largely similar to the non-public-facing EW model, except that infections associated with hospital care of infected individuals become a major source of infection for healthcare EWs.

Thus it is a fundamental aspect of all three EW models that one important and potentially observable feature of the pandemic—who is infecting who—is expected to change over time.

## Discussion

It is intuitive that, when populations are sheltering in place to reduce virus transmission, EWs whose jobs require them to maintain contacts with each other and/or the public have a higher risk of infection, and, if infected, an increased probability of spreading. Yet the precise nature of this effect has received relatively little attention in COVID-19 modeling and public planning. Our goal here was to address a critical issue, how variation in the contact profiles of common types of EWs affect their disease risk and efforts to control the pandemic, within the context of the epidemiological models that are widely used to guide policy decisions.

We emphasize that, while our modeling leads to several general observations about the potential effects of EWs on the pandemic dynamics, especially for COVID-19, it was not designed to predict pandemic progress in specific populations. Fitting models including essential workers is all the more challenging, since existing data on essential workers is often limited to cross-sectional measurements (e.g., infection prevalence on a single day, total deaths, or seroprevalence) compared between essential (or frontline) workers and others [15–17, 20–22] and is confounded by correlated factors (e.g., neighborhood location, socioeconomic status, family size, and race/ethnicity) [15, 49]. Moving towards predictive models would require, at a minimum, accurate data on contacts among EWs and between EWs and nEWs in the specific context being modeled, incorporation of demographic differences in EW and nEW subpopulations, and treatment of the compartmentalization of both subpopulations into specific workplaces and households [11, 50–52].

Moreover, the $R_0$ value we used in the model may be more representative of urban areas, which, because of population density, public transportation and other factors, tend to have higher rates of contact than do regions with lower population density [53]. Nonetheless, empirical data on rates of infection and fatalities in groups of EWs in urban areas hard-hit by COVID-19 paint a clear picture of increased individual risk [54–57], and the models developed here suggest that, at a minimum, this is likely to be a pervasive challenge in population centers. There is also evidence for elevated EW risk in factories and food-processing facilities in areas with lower population density [58]; however we did not explore how the various models behave in such conditions. Additionally, we treat all parameters as fixed over time, ignoring the emergence of new viral variants with higher $R_0$ or differences in other epidemiological parameters [59].

We also have not fully explored the full range of scenarios that our models could capture. In particular, we focused on the impact on and of EWs within the context of SIP orders applied in the midst of a rapidly growing local outbreak, as occurred repeatedly across the planet in the first few months of the COVID-19 pandemic. EWs may have a very different disease risk and impact on the epidemic when SIP is applied before a large number of individuals are infected, or after the outbreak is much farther along or already under control. There is also likely to be a

significant effect of the conditions under which EWs under consideration live and work (e.g., [60]). Our models, for example, do not take into account protective measures taken by EWs, in particular healthcare workers, which are likely to ameliorate infectious transmission among EWs and between them and nEWs (notably for public-facing EWs).

These limitations notwithstanding, it is clear that the type of essential work in which a person is engaged has a big effect on their individual and collective risk of infection. For example, even with limited exposure to the public, EWs at high interaction workplaces such as manufacturing and food processing facilities, or with high exposure to infected individuals, are at the highest individual risk of infection. But more public-facing workers, such as cashiers, even when they have a much lower individual risk, can have a much greater impact on the pandemic.

The specific observations depend on our modeling assumptions and parameters, yet these results highlight the importance of not treating EWs as an undifferentiated class. Our model established the extent of EW interaction with nEWs as a critical feature in studying the effects of SIP orders and EWs on individual and collective risk. Similar to models that include separate compartments to differentiate populations by age or neighborhood, we should also differentiate subpopulations that have fundamentally distinct contact profiles, such as EW versus nEWs. Including essential workers can have substantial effects on the predicted number of infections and deaths from COVID-19 [61]. While models that account for heterogeneity in disease spread [14] capture some features of our model, understanding the effects of specific types of workers and any related interventions requires that essential workers be modeled explicitly.

Although our model is parameterized using data from COVID-19 infections, we expect our qualitative findings to generalize to similar future pandemics. The success of SIP orders and social distancing in slowing COVID-19 transmission [62, 63] suggests that they are likely to be used as tools for controlling future pandemics. However, all SIP orders require a substantial fraction of the population be exempt from SIP to maintain essential services. How many and which workers should be exempt from SIP orders deserves further consideration and will depend on characteristics of the virus (e.g., on $R_0$).

In that regard, our model makes clear that policy decisions should consider both the collective and individual risk associated with the numbers and contact profiles of different types of EWs. For example, a larger pool of EWs requires that the remainder of the population shelter in place more stringently to maintain pandemic suppression. EWs in greater contact with the much larger nEW subpopulation contribute more to collective risk and thus require greater increases in stringency. Policy decisions regarding EWs must consider the willingness and ability of nEWs to increase the stringency of SIP.

Alternatively, policy decisions could mitigate the contact profiles, and therefore disease transmission, of EWs. While we treat these profiles as static, they are not: the deployment of personal protective equipment, workplace social distancing policies, and vaccines in both EWs and nEWs can play a significant role in limiting EW exposure to infected individuals, and, if infected, minimizing their role in disease transmission [64]. Widespread workplace testing for infections and temporary removal of infected individuals from the workplace would also reduce transmission. Modeling efforts like ours can help inform the best way to target these interventions, including optimal vaccine allocation (e.g., [65–67]).

## Modeling details

We were guided by two principles in designing our models for different types of EWs. First, we wanted them to capture essential characteristics of archetypes of EWs: cashiers interact

extensively with the public, factory workers and healthcare workers have a large fraction of their contacts in the workplace, and healthcare workers are exposed to a unique infection risk from their exposure to a high concentration of infected individuals. Second, we wanted the models to be simple enough that we could connect the modeling results back to these essential characteristics. Hence none of these models should be considered to fully represent a real group of EWs; rather they represent distinct essential characteristics that are often found in real groups of EWs.

All models are described in terms of the number of potentially infectious contacts per day, where these contacts are apportioned between subpopulations according to $\beta_{ij}$, which represents the number of contacts an individual in subpopulation $i$ has with individuals in subpopulation $j$. Representing EWs by $E$ and nEWs by $N$, the four corresponding parameters are $\beta_{NN}, \beta_{NE}, \beta_{EN}$, and $\beta_{EE}$. We assume that contacts are symmetric, such that with proportion $f$ of EWs and $1-f$ of NEs:

$$(1-f)\beta_{NE} = f\beta_{EN}.$$

We further assume that *before SIP* every individual has a set total number of contacts $\beta_T$ such that:

$$\beta_{NE} + \beta_{NN} = \beta_{EE} + \beta_{EN} = \beta_T.$$

For public-facing EWs, we assumed that before SIP contacts between subpopulations are proportional to the susceptible subpopulation's size such that:

$$\beta_{EE} = f\beta_T, \ \beta_{EN} = (1-f)\beta_T, \ \beta_{NE} = f\beta_T, \text{ and } \beta_{NN} = (1-f)\beta_T.$$

As we are further assuming that public-facing EWs contacts are unaffected by SIP, $\theta$ is applied only to $\beta_{NN}$, such that after SIP the parameters are:

$$\beta_{EE} = f\beta_T, \ \beta_{EN} = (1-f)\beta_T, \ \beta_{NE} = f\beta_T, \text{ and } \beta_{NN} = \theta(1-f)\beta_T.$$

For non-public-facing and healthcare EWs, prior to the proportional assortment of contacts, we reserve a fraction $\rho$ of their contacts to be with other EWs. Thus, we assume that before SIP:

$$\beta_{EE} = (\rho + f(1-\rho))\beta_T, \ \beta_{EN} = (1-f)(1-\rho)\beta_T, \ \beta_{NE} = f(1-\rho)\beta_T, \text{ and}$$

$$\beta_{NN} = (1-f(1-\rho))\beta_T.$$

We further assume that all contacts except the $\rho\beta_T$ reserved to be among EWs in the workplace are reduced by SIP, and thus post SIP the parameters are:

$$\beta_{EE} = (\rho + \theta f(1-\rho))\beta_T, \ \beta_{EN} = \theta(1-f)(1-\rho)\beta_T, \ \beta_{NE} = \theta f(1-\rho)\beta_T, \text{ and}$$

$$\beta_{NN} = \theta(1-f(1-\rho))\beta_T.$$

For the examples presented here, we set $\rho = 0.5$. We show results with $\rho = 0$ in Figs 10, 23, and 40 in S1 Appendix.

The healthcare model builds on the non-public facing model, but in this case all hospitalized individuals have an extra $\beta_{HE}$ contacts with healthcare workers per day. We assume $\beta_{HE}$ scales with the number of healthcare workers, and parameterize the choice of its value such that a healthcare worker is $\kappa$ times more likely to get infected by a nEW than is a nEW (see Section 5 of S1 Appendix for details). In the main text, we set $\kappa = 1.5$. Alternative parameter values and parameterization of $\beta_{HE}$ are explored in Section 5 of S1 Appendix.

The other parameters used in the SEIR model come largely from [6, 7], as detailed in Table 1, with further consideration of the dynamics of COVID-19 in New York City before SIP. In particular, we assume that $R_0 = 3$, which is consistent with previous estimates (see Table 1), and rely on $R_0$ to determine the value of $\beta_T$ given values of the other parameters. We further placed the start of SIP at day 53, such that our model without EWs predicts approximately the observed number of deaths in New York City on the day formal SIP orders were issued. This is relatively late in the pandemic progression, in that a substantial fraction of the population was already infected (as seems to have been the case in New York City, according to recent serological estimates [68, 69]), and accounts for the high prevalences in the models. In S1 Appendix, we present results for alternative parameter values, which are qualitatively—and often quantitatively—similar.

## Supporting information

**S1 Appendix.**
(PDF)

## Acknowledgments

We thank Laura Hayward, Magnus Nordborg, Michael Zeitz and two anonymous reviewers for helpful discussions or comments on the manuscript.

## Author Contributions

**Conceptualization:** William R. Milligan, Zachary L. Fuller, Ipsita Agarwal, Michael B. Eisen, Molly Przeworski, Guy Sella.

**Data curation:** Zachary L. Fuller.

**Investigation:** William R. Milligan, Zachary L. Fuller, Ipsita Agarwal.

**Methodology:** William R. Milligan, Zachary L. Fuller, Michael B. Eisen, Molly Przeworski, Guy Sella.

**Software:** William R. Milligan, Zachary L. Fuller.

**Supervision:** Michael B. Eisen, Molly Przeworski, Guy Sella.

**Visualization:** William R. Milligan, Zachary L. Fuller.

**Writing – original draft:** William R. Milligan, Zachary L. Fuller, Ipsita Agarwal, Michael B. Eisen, Molly Przeworski, Guy Sella.

**Writing – review & editing:** William R. Milligan, Zachary L. Fuller, Ipsita Agarwal, Michael B. Eisen, Molly Przeworski, Guy Sella.

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
