## [Decision Letter · Decision Letter 0]

9 Apr 2021

PONE-D-21-02147

Impact of essential workers in the context of social distancing for epidemic control

PLOS ONE

Dear Dr. Milligan,

Thank you for submitting your manuscript to PLOS ONE. After careful consideration, we feel that it has merit but does not fully meet PLOS ONE’s publication criteria as it currently stands. Therefore, we invite you to submit a revised version of the manuscript that addresses the points raised during the review process.

We look forward to receiving your revised manuscript.

Kind regards,

Chiara Poletto

Academic Editor

PLOS ONE

Journal Requirements:

Reviewers' comments:

Reviewer's Responses to Questions

**Comments to the Author**

1. Is the manuscript technically sound, and do the data support the conclusions?

Reviewer #1: Yes

Reviewer #2: Yes

2. Has the statistical analysis been performed appropriately and rigorously? 

Reviewer #1: Yes

Reviewer #2: N/A

3. Have the authors made all data underlying the findings in their manuscript fully available?

Reviewer #1: Yes

Reviewer #2: Yes

4. Is the manuscript presented in an intelligible fashion and written in standard English?

Reviewer #1: Yes

Reviewer #2: Yes

5. Review Comments to the Author

Reviewer #1: Overview and strengths

This study used a compartmental SEIR model to investigate the role that essential workers versus non-essential worker play in spreading SARS-CoV-2 before and after a shelter in place order. The modeling work used an “archetype” framework, where different types of essential works were identified at different risk levels based on whether they were public-facing, not public-facing, or healthcare workers. This was an interesting and novel approach to consider differences in essential worker job functions and how those may affect SARS-CoV-2 spread. Additionally, the writing was clear and concise, and the manuscript was pleasantly easy to read.

Major comments

-Choices about specific parameter values were not clear to me. For example, for t_IR, why did you choose 5 days if literature estimates ranged from 5 days to 2 weeks? For t_CR, the literature estimate was 10 days, but you chose 7. Why? I think it is necessary to justify these decisions a bit more. Further, given there is such variability in the parameter values from different studies, it would be nice to consider a sweep across plausible values. Are your results sensitive to difference in these values? Or are conclusions robust to differences in parameter values? This seems like an open question that needs to be considered. Also, I couldn’t figure out the actual R0 you considered. I see the different R0’s relative to relative success of SIP, but what does that actually correspond to?

-In the figure captions there are varying degrees of context, which makes it a bit hard to know what to expect from the caption. For example, in Figure 3, I was having trouble interpreting the graph (e.g., why one plot had a red line but others did not). Some of that information on how to interpret it was listed in the text (lines 233-237), but it might be nice to have that directly in the caption. On the flip side, for Figure 4, there was a long paragraph explaining both the context of the figure and the conclusions you were drawing from that figure. This was helpful, but it seemed strange when compared to Figure 3. I suggest an intermediary, some context is helpful, but some could be left to the results itself.

-I think the archetype aspect of the study is unique and interesting and should be highlighted a bit more. I think the authors should consider a conceptual figure that more concisely describes the different archetypes and gives more examples of each. Where would farmers fall versus factory workers? What are the defining characteristics of each? You could have a figure with icons for different types of workers that might be in each category.

Minor comments

-Line 190-191: Do these reductions in R0 compare to some meaningful reduction? Percent reduction in R0? Might be good to include a citation here.

-Line 190: Does 5% of the population as EWs correspond to some data? At different points in the epidemic, different people were considered “essential,” does this only refer to those that were considered essential at the very beginning of the pandemic? Where would say hair dressers or nail stylists fall in here? They aren’t essential, but they were one of the first to go back to work, and are surely getting exposed a lot. No need to classify every single profession, but just curious how you’re defining the timeline of EW and where the 5% comes from.

-Line 206-208: I think it is ok to not take those into account, but perhaps you could speak to (in the discussion maybe) what it would look like if you did take these into account.

-Line 256-257: It is not clear to me why this would yield an intuitive exploration of dynamics between and within EWs and nEWs. Could you please add a bit more of an explanation?

-Line 265-266: sentence that spans these two lines has a grammatical issue

-Figure 2: It is difficult to see the no EWs line. I am also confused about the solid colored lines. Are these individuals in those archetypes that weren’t essential workers? I didn’t realize that was being considered in this analysis. Please clarify and update figure or manuscript text to better explain this.

Reviewer #2: Summary

Milligan et al. present a simple theoretical analysis of the role different types of idealized workers play in transmission of SARS-CoV-2 while other members of society socially distance. The models are SEIR-type models parameterized with literature values rather than calibrated to data, and so may preclude any specific or quantitative analyses. Nonetheless, the authors are able to establish several general qualitative principles, the manuscript is on the whole well written, enjoyable to read, and easy to follow, and the authors are clear that the model is not intended to provide quantitative predictions. I only have minor concerns, and these pertain mainly to inconsistent and ambiguous use of terminology and two suggests for supplementary figures. It would also be worth updating several aspects of the introduction and discussion to discuss important changes since the initial submission and how these may affect results, particularly relating to widespread mask-wearing and vaccination. See below for specific comments.

Abstract

• Line 25: you could say ‘transmission model’ or ‘mathematical model’ so it’s clear to the reader that this is not just a conceptual model.

Modeling Essential Workers

• I think the length of the infectious period for those in critical care is too long, if I am understanding your methodology correctly. It seems like people’s infectiousness peaks early, just before symptom onset, and is very low by 10 days following infection (see e.g. https://www.nature.com/articles/s41467-021-21710-6 or https://www.nature.com/articles/s41591-020-0869-5). This impacts significantly your third model, the HCW one. It might be worth exploring the effect of turning infectiousness off, or reducing it, for part of the critical care compartments, showing the results in the supplementary material, and referring to them in the main text Results section. At the very least you should describe how a different assumption about the infectious period might affect your results for model C in the discussion.

• I also think your assumption about latent period might be too long (see the above mentioned references). The incubation period is on average 5 days, but people are infectious several days before that. It could also be worth doing a supplementary figure with a short latent period, or at least discussing what impact it would have. I imagine it would mainly be on the speed with which public facing essential workers come to be the dominant source of transmission in Fig. 4C?

• Line 171-173 – why do you make the assumption that half their contacts are with other EWs? Why do you only make this assumption for the non-public facing EWs?

• Line 179: I recommend you write ‘contact’ rather than ‘interaction’ here, and in all other places were you use interaction to mean contact, to avoid confusion with statistical interaction.

• Fig. 2 might be hard to read for a color-blind person. It might be worth checking that your color-palette is color-blind friendly and if not using a color-palette that is, like viridis.

• Line 206, the term ‘rate of infection’ is ambiguous here and it’s not clear what you mean. If I understand you correctly, I think a better term is ‘infection attack rate’

Results

• Line 224-225. Sorry if this seems pedantic, but I’m not sure ‘take root’ is a good way to phrase this, as infections are still occurring in the nEW group in your model. Perhaps you could just say ‘those infections lead to little onward transmission in the nEW group, due to SIP’?

• Fig. 4C: please describe the acronyms in the legend. In particular, I don’t know what WH means.

• Lines 287: do you mean incidence here, or prevalence on line 286? Here and in other places you seem to be using prevalence and incidence interchangeably. Please check all uses of these terms and ensure you are using the correct one each time.

• Line 288: Rather than ‘tracks with’, can you say ‘decays at the same rate’, or something like that?

Discussion

• Could you discuss how widespread mask use among both nEWs and EWs would affect your results?

• Could you discuss what implications your results have, if any, on vaccine prioritization? There are also several modeling preprints which attempt to assess prioritization strategies which I think do include EWs – it would be nice to compare your results to theirs here. https://www.ncbi.nlm.nih.gov/pmc/articles/PMC7523157.2/
https://www.medrxiv.org/content/10.1101/2021.02.23.21252309v1
https://www.medrxiv.org/content/10.1101/2021.03.04.21251264v1

(there may be others – I just did a quick search on google scholar and these were the first that came up).

• Line 347, the word ‘different’ is missing

6. PLOS authors have the option to publish the peer review history of their article (what does this mean?). If published, this will include your full peer review and any attached files.

Reviewer #1: No

Reviewer #2: No

---

## [Author Response · Author response to Decision Letter 0]

1 Jun 2021

We thank the reviewers for their thoughtful and helpful comments, which we believe we have fully addressed, as detailed in the file "Response to Reviewers" and reproduced below:

Reviewer #1: Overview and strengths

This study used a compartmental SEIR model to investigate the role that essential workers versus non-essential workers play in spreading SARS-CoV-2 before and after a shelter in place order. The modeling work used an “archetype” framework, where different types of essential works were identified at different risk levels based on whether they were public-facing, not public-facing, or healthcare workers. This was an interesting and novel approach to consider differences in essential worker job functions and how those may affect SARS-CoV-2 spread. Additionally, the writing was clear and concise, and the manuscript was pleasantly easy to read.

We appreciate the reviewer’s kind assessment.

Major comments

-Choices about specific parameter values were not clear to me. For example, for t_IR, why did you choose 5 days if literature estimates ranged from 5 days to 2 weeks? For t_CR, the literature estimate was 10 days, but you chose 7. Why? I think it is necessary to justify these decisions a bit more. 

We agree that some of our choices required further clarification and have now added this information in the caption of Table 1 and more citations to justify them. Specifically:

While estimates of tIR vary between 5 days to 2 weeks, 5 days appears to be the commonly accepted value (see citations in Table 1). 

For tCR, we split the estimate of 10 days into 7 days in critical care and 3 days in the hospital after critical care. 

We now use tE=3 to account for presymptomatic transmission.

We now set R0=3, which agrees with previous estimates (see citations in Table 1).

Further, given there is such variability in the parameter values from different studies, it would be nice to consider a sweep across plausible values. Are your results sensitive to differences in these values? Or are conclusions robust to differences in parameter values? This seems like an open question that needs to be considered.

We expanded our sensitivity analyses and now devote a chapter in the Appendix to it. These analyses are noted where we describe our parameter choices (lines 121-124): “In S1 Appendix, we vary parameter values to check that our qualitative results are insensitive to variation within the range examined (See Figs. S1-16 for alternative versions of Fig, 2, Figs S17-29 for Fig. 3, and Figs. S30 - 46 for Fig. 4)” and we refer the reader to specific appendix figures at the end of each figure caption that relies on our parameter choices. As we note in the above excerpt, our qualitative results are robust to the variations in parameters examined.

Also, I couldn’t figure out the actual R0 you considered. I see the different R0’s relative to relative success of SIP, but what does that actually correspond to?

We apologize for the lack of clarity as to value of R0. We now note in line 475-476 that: “In particular, we assume that R0=3, which is consistent with previous estimates (see Table 1)...””and have added the new chosen value, R0=3 , in Table 1, alongside relevant references; we also explore alternative values of R0 in S1 Appendix. 

-In the figure captions there are varying degrees of context, which makes it a bit hard to know what to expect from the caption. For example, in Figure 3, I was having trouble interpreting the graph (e.g., why one plot had a red line but others did not). Some of that information on how to interpret it was listed in the text (lines 233-237), but it might be nice to have that directly in the caption. On the flip side, for Figure 4, there was a long paragraph explaining both the context of the figure and the conclusions you were drawing from that figure. This was helpful, but it seemed strange when compared to Figure 3. I suggest an intermediary, some context is helpful, but some could be left to the results itself.

We thank the reviewer for bringing this point to our attention. We have now added all necessary information to understand figures in the captions and moved interpretation of figures to the main text. In particular, the lack of red contour lines in Fig. 3 is now explained in the caption and the interpretation previously provided in the caption of Fig. 4 has now been moved to the main text. 

-I think the archetype aspect of the study is unique and interesting and should be highlighted a bit more. I think the authors should consider a conceptual figure that more concisely describes the different archetypes and gives more examples of each. Where would farmers fall versus factory workers? What are the defining characteristics of each? You could have a figure with icons for different types of workers that might be in each category.

We appreciate the suggestion of a conceptual figure. Figure 1B-D goes some way in that direction by showing the contacts affected by SIP for each archetype. Additionally, we provide concise explanations of the basic characteristics of each archetype along with examples (lines 169-170, 176, and 187).

Minor comments

-Line 190-191: Do these reductions in R0 compare to some meaningful reduction? Percent reduction in R0? Might be good to include a citation here.

We added the values of , the proportion of R0 that remains, for each value of R0(lines 206-208): “ values corresponding to partially effective (=0.5 and R0=1.5), effective (R0=0.9 and =0.3) and highly effective (R0=0.5 and =0.16) SIP”; the relative reduction is 1-. 

-Line 190: Does 5% of the population as EWs correspond to some data? 

As we note in the introduction (lines 59 - 62): “In New York City, workers in categories deemed essential (as of March 2020 [10]) are estimated to comprise a quarter of the workforce [11], or over 1M people, of whom over half are employed in healthcare and 15% in grocery, convenience and drug stores. Estimates in California are that one in eight individuals is considered an essential worker [12].”. With a total population of NYC of ~8.4 M, this suggests that healthcare workers compose ~6% and that grocery, convenience and drug store workers compose about ~2%, hence our choice of 5%. For sake of comparison, we used the same value of 5% for all archetypes. Additionally, in figure 3, we show how different numbers of EWs could affect the dynamics, and in S1 Appendix Figs. S1 and S32 we consider the case of f=0.1. 

At different points in the epidemic, different people were considered “essential,” does this only refer to those that were considered essential at the very beginning of the pandemic? 

While we were thinking about the time at which stringent SIP measures were implemented, some qualitative results may apply at different stages.

Where would say hair dressers or nail stylists fall in here? They aren’t essential, but they were one of the first to go back to work, and are surely getting exposed a lot. No need to classify every single profession, but just curious how you’re defining the timeline of EW and where the 5% comes from.

As far as we know, hair dressers were not considered essential during SIP, so we feel it would be confusing to include them here. However, one could imagine a version of public-facing workers at later stages fitting this profession as well. In that regard, the rough figure of 5%, chosen as a basis for comparison among different archetypes, could be considered to include them at a later stage than the initial SIP.

-Line 206-208: I think it is ok to not take those into account, but perhaps you could speak to (in the discussion maybe) what it would look like if you did take these into account.

In the Discussion, we write that (lines 383 - 385): “Our models, for example, do not take into account protective measures taken by EWs, in particular healthcare workers, which are likely to ameliorate infectious transmission among EWs and between them and nEWs (notably for public-facing EWs)”. The bolded portion corresponds to the revised text. A detailed investigation of these effects is beyond the scope of this paper, but we agree that it may be of interest in follow-up work.

-Line 256-257: It is not clear to me why this would yield an intuitive exploration of dynamics between and within EWs and nEWs. Could you please add a bit more of an explanation?

Depicting the time-resolved curves of the model using the same parameters for all archetypes allowed us to better understand how differences between archetypes (as opposed to differences in parameter values) affect the results. We chose f=0.05for the reasons noted above, and R0 =0.5 as an example with stringent social distancing. Additionally, choosing substantially larger values of R0 values would result in quickly reaching herd immunity. In S1 Appendix Figs. S30 and S31, we show how the results of figure 4 with different values of R0 .

-Line 265-266: sentence that spans these two lines has a grammatical issue

We thank the reviewer for catching this point; we have revised this sentence.

-Figure 2: It is difficult to see the no EWs line. I am also confused about the solid colored lines. Are these individuals in those archetypes that weren’t essential workers? I didn’t realize that was being considered in this analysis. Please clarify and update figure or manuscript text to better explain this.

We made the EWs line more visible. The solid lines refer to the cumulative number of infections within the non-essential worker sub-population (which constitute proportion1-f of the population), whereas the dashed lines refer to the cumulative infections within the essential worker sub-population. We now clarify this point in the figure caption.

Reviewer #2: Summary

Milligan et al. present a simple theoretical analysis of the role different types of idealized workers play in transmission of SARS-CoV-2 while other members of society socially distance. The models are SEIR-type models parameterized with literature values rather than calibrated to data, and so may preclude any specific or quantitative analyses. Nonetheless, the authors are able to establish several general qualitative principles, the manuscript is on the whole well written, enjoyable to read, and easy to follow, and the authors are clear that the model is not intended to provide quantitative predictions. I only have minor concerns, and these pertain mainly to inconsistent and ambiguous use of terminology and two suggestions for supplementary figures. It would also be worth updating several aspects of the introduction and discussion to discuss important changes since the initial submission and how these may affect results, particularly relating to widespread mask-wearing and vaccination. See below for specific comments.

We appreciate the reviewer’s kind assessment.

Abstract

• Line 25: you could say ‘transmission model’ or ‘mathematical model’ so it’s clear to the reader that this is not just a conceptual model.

Thank you. We updated the text accordingly.

Modeling Essential Workers

• I think the length of the infectious period for those in critical care is too long, if I am understanding your methodology correctly. It seems like people’s infectiousness peaks early, just before symptom onset, and is very low by 10 days following infection (see e.g. https://www.nature.com/articles/s41467-021-21710-6 or https://www.nature.com/articles/s41591-020-0869-5). This impacts significantly your third model, the HCW one. It might be worth exploring the effect of turning infectiousness off, or reducing it, for part of the critical care compartments, showing the results in the supplementary material, and referring to them in the main text Results section. At the very least you should describe how a different assumption about the infectious period might affect your results for model C in the discussion.

Following the reviewer’s suggestion, in Figs. S16, S29, and S46 of Appendix S1, we explored how assuming that individuals in critical care (or that have recovered from critical care) are not infectious affected our results. The quantitative results remain similar, and the qualitative conclusions--that healthcare workers are at higher personal risk but have lower impacts on the broader pandemic than public-facing workers--did not change. We also see the same if we change the amount of time spent in the hospitalized compartments (HR and HC) or in the infected compartments (IR,IA,IH) (Figs. S14-15, S27-28, and S44-45 in Appendix S1). 

• I also think your assumption about latent period might be too long (see the above mentioned references). The incubation period is on average 5 days, but people are infectious several days before that. It could also be worth doing a supplementary figure with a short latent period, or at least discussing what impact it would have. I imagine it would mainly be on the speed with which public-facing essential workers come to be the dominant source of transmission in Fig. 4C?

We appreciate the input. We changed the latency period to 3 days to account for pre-symptomatic transmission. We also include supplementary figures where we alter the latency period by 1 day (Figs. S7-8, S20-21, S37-38 in Appendix S1). 

Line 171-173 – why do you make the assumption that half their contacts are with other EWs? Why do you only make this assumption for the non-public facing EWs?

We added the following clarification (lines 181-183): “These features are meant to account for frequent and close contacts in the workplace, and may be essential to explaining outbreaks within, say, factories [47,48]”.

In addition, we also now clarify that we make a similar assumption in healthcare EWs model (lines 192-195): “Similar to the non-public-facing essential worker model, we assume half of healthcare workers’ contacts are with other healthcare workers because of their close contact at the workplace (see Figs. S10, S23, and S40 in Appendix S1 for results with proportionate mixing)”. 

The qualitative effects of this assumption become apparent when we compare the results of the non-public-facing model to those of a similar model without it, i.e., =0 (Fig. S10-11, S20-21, S40-41 in Appendix S1). Without this assumption, the effects of EWs are minimal, as only 2.5% of contacts are not affected by SIP. For completeness, in Appendix S1, we show the results of all models assuming half of EW contacts are with other EWs or that all contacts are distributed via proportionate mixing. Setting =0.5 for public-facing EWs leads EWs to be at greater personal risk of infection but to have smaller impacts on the broader pandemic. Alternatively, setting =0 for non-public-facing and healthcare EWs decreases their personal risk of infection but increases their effect on collective risk. 

• Line 179: I recommend you write ‘contact’ rather than ‘interaction’ here, and in all other places where you use interaction to mean contact, to avoid confusion with statistical interaction.

We have updated the terminology accordingly. 

• Fig. 2 might be hard to read for a color-blind person. It might be worth checking that your color-palette is color-blind friendly and if not using a color-palette that is, like viridis.

We apologize for this oversight and have updated our figures accordingly.

• Line 206, the term ‘rate of infection’ is ambiguous here and it’s not clear what you mean. If I understand you correctly, I think a better term is ‘infection attack rate’

We changed it to “infection prevalence”.

Results

• Line 224-225. Sorry if this seems pedantic, but I’m not sure ‘take root’ is a good way to phrase this, as infections are still occurring in the nEW group in your model. Perhaps you could just say ‘those infections lead to little onward transmission in the nEW group, due to SIP’?

We have revised the sentence as suggested (lines 240-242). 

• Fig. 4C: please describe the acronyms in the legend. In particular, I don’t know what WH means.

We appreciate the catch. The caption now states: “The acronyms denote different types of transmission, with the first and second letters denoting the infecting and the infected subpopulation respectively (e.g., EN is infections from EWs to nEWs); WH refers to infections that occur within hospitals”.

 • Lines 287: do you mean incidence here, or prevalence on line 286? Here and in other places you seem to be using prevalence and incidence interchangeably. Please check all uses of these terms and ensure you are using the correct one each time.

We thank the reviewer for bringing this point to our attention. We have now reviewed the manuscript to make sure that we use the terminology correctly and consistently. 

• Line 288: Rather than ‘tracks with’, can you say ‘decays at the same rate’, or something like that?

We changed “tracks with” to “decays at the same rate as”.

Discussion

• Could you discuss how widespread mask use among both nEWs and EWs would affect your results?

We now write (lines 383-385): “Our models, for example, do not take into account protective measures taken by EWs, in particular healthcare workers, which are likely to ameliorate infectious transmission among EWs and between them and nEWs (notably for public-facing EWs)”. The bolded portion corresponds to the revised text. A detailed investigation of these effects is beyond our scope here, but we agree that it may be of interest in follow up work.

• Could you discuss what implications your results have, if any, on vaccine prioritization? There are also several modeling preprints which attempt to assess prioritization strategies which I think do include EWs – it would be nice to compare your results to theirs here. https://www.ncbi.nlm.nih.gov/pmc/articles/PMC7523157.2/
https://www.medrxiv.org/content/10.1101/2021.02.23.21252309v1
https://www.medrxiv.org/content/10.1101/2021.03.04.21251264v1

(there may be others – I just did a quick search on google scholar and these were the first that came up).

We appreciate the importance of this point. However, given that our model is meant as illustrative of the importance of EWs on the pandemic dynamics and not as a basis for inference or recommendations, we would rather steer away from drawing any prescriptive implications. We have included the citations you provided when we write (lines 427-428) “Modeling efforts like ours can help inform the best way to target these interventions, including optimal vaccine allocation (e.g., [65–67])”.

Line 347, the word ‘different’ is missing

We have revised the sentence accordingly.

---

## [Decision Letter · Decision Letter 1]

22 Jul 2021

Impact of essential workers in the context of social distancing for epidemic control

PONE-D-21-02147R1

Dear Dr. Milligan,

We’re pleased to inform you that your manuscript has been judged scientifically suitable for publication and will be formally accepted for publication once it meets all outstanding technical requirements.

Kind regards,

Chiara Poletto

Academic Editor

PLOS ONE

Additional Editor Comments (optional):

Reviewers' comments:

Reviewer's Responses to Questions

**Comments to the Author**

1. If the authors have adequately addressed your comments raised in a previous round of review and you feel that this manuscript is now acceptable for publication, you may indicate that here to bypass the “Comments to the Author” section, enter your conflict of interest statement in the “Confidential to Editor” section, and submit your "Accept" recommendation.

Reviewer #2: All comments have been addressed

2. Is the manuscript technically sound, and do the data support the conclusions?

Reviewer #2: Yes

3. Has the statistical analysis been performed appropriately and rigorously? 

Reviewer #2: Yes

4. Have the authors made all data underlying the findings in their manuscript fully available?

Reviewer #2: Yes

5. Is the manuscript presented in an intelligible fashion and written in standard English?

Reviewer #2: Yes

6. Review Comments to the Author

Reviewer #2: (No Response)

7. PLOS authors have the option to publish the peer review history of their article (what does this mean?). If published, this will include your full peer review and any attached files.

Reviewer #2: No

---

## [Editor Report · Acceptance letter]

27 Jul 2021

PONE-D-21-02147R1 

Impact of essential workers in the context of social distancing for epidemic control 

Dear Dr. Milligan:

I'm pleased to inform you that your manuscript has been deemed suitable for publication in PLOS ONE. Congratulations! Your manuscript is now with our production department. 

Kind regards, 

on behalf of

Dr. Chiara Poletto 

Academic Editor

PLOS ONE